# Quality of Life Changes in Early-Onset Multiple Sclerosis: A 4-Year Follow-Up Study

**DOI:** 10.3390/jcm11175226

**Published:** 2022-09-04

**Authors:** Laura Rosa, Maria Petracca, Antonio Carotenuto, Pasquale Dolce, Kyrie Piscopo, Francesca Dicé, Francesca Lauro, Antonio Luca Spiezia, Marcello Moccia, Luigi Lavorgna, Carmine Iacovazzo, Giuseppe Servillo, Nelson Mauro Maldonato, Alessandro Chiodi, Vincenzo Brescia Morra, Roberta Lanzillo

**Affiliations:** 1Department of Public Health, University of Naples Federico II, 80131 Naples, Italy; 2Department of Neurosciences and Reproductive and Odontostomatological Sciences, University of Naples Federico II, 80131 Naples, Italy; 3Department of Human Neurosciences, Sapienza University, 00185 Rome, Italy; 4SInAPSi University Centre, University of Naples Federico II, 80138 Naples, Italy; 5Clinical Psychology Unit, ASST Santi Paolo e Carlo, 20142 Milan, Italy; 6Department of Human Studies, University of Naples Federico II, 80138 Naples, Italy; 7Division of Neurology, Department of Advanced Medical and Surgical Sciences, AOU-University of Campania “Luigi Vanvitelli”, 80138 Naples, Italy; 8Intradepartmental Program of Clinical Psychology, Federico II University Hospital, 80131 Naples, Italy

**Keywords:** multiple sclerosis, quality of life, early-onset

## Abstract

This study investigates longitudinal changes in health-related quality of life (HRQoL) in early-onset multiple sclerosis (MS) patients and explores the impact of disease activity (relapses) on such changes. People with MS (PwMS) onset between 12 and 25 years of age were followed longitudinally. At baseline and at year 4, patients were asked to fill the Paediatric Quality of life inventory (PedsQL). Demographic and clinical features were collected at both time points. Longitudinal within-group comparison of HRQoL total score and sub-scores was performed via paired samples t-test. The effect of relapses on the HRQoL changes over time was explored via linear mixed-effects analysis. No longitudinal changes were observed in the overall PedsQL score, nor in the physical, school and psychological functioning. An increase in the social functioning subscale (*p* < 0.001) and a decrease in the emotional subscale (*p* = 0.006) were observed. The change in social functioning, but not the one in the emotional subscale, was affected by the occurrence of relapses (*p* = 0.044). In conclusion, stimulating the patients to accept their emotional responses to health-related limitations, while preserving their social and relational resources seems key to the preservation of an adequate QoL over time in juvenile-onset MS.

## 1. Introduction

Multiple sclerosis (MS) is a chronic autoimmune disease of the central nervous system, representing the first cause of non-traumatic motor and cognitive disability in young adults [1]. As such, it has a strong impact on patients′ health-related quality of life (HRQoL) [2]. HRQoL can be defined as the individual perception of one’s wellness, and it relies on daily functioning and on how physical, psychological and social factors influence the daily activities of a person [3].

HRQoL perception is not static along the disease course and shows a complex relationship with other aspects of the disease, such as physical disability and mental and psychosocial functioning. In adult-onset MS, longitudinal studies have shown that, notwithstanding the worsening of physical function, social, mental and psychosocial functioning tends to become stable or improve over time [4,5]. However, most studies about MS HRQoL have been cross-sectional [6,7], and the few longitudinal studies have not accounted for the potential heterogeneity of young patients with MS [6,7].

In early-onset MS (onset age ≤ 25 years) cross-sectional studies have shown that HRQoL is influenced by sex, race, disability, disease duration, fatigue and time to disability accrual [6,7,8], but longitudinal analyses exploring HRQoL evolution over time are lacking. Moreover, cognitive aspects and their impact on HRQoL should be investigated appropriately, and some efforts have been made, such as a newly designed brief and easy-to-use screening test to help to identify cognitive impairment, fatigue, and loss of HRQoL early in patients with pediatric onset MS [9].

Additionally, the impact of specific aspects, peculiar to this population, on HRQoL has never been explored. Indeed, emotional and psychological elements consequent to the communication of the diagnosis in a transitional stage such as adolescence might have a long-term effect on patients′ HRQoL. Moreover, high inflammatory activity, which characterizes pediatric forms of MS with frequent occurrence of relapses [10], is likely to affect the patients′ perceived HRQoL. Given this background, the objective of this study was to conduct a longitudinal analysis (i) investigating HRQoL changes in early-onset MS over 4 years and (ii) explore the impact of clinical activity and progression on such changes.

## 2. Methods

### 2.1. Procedures and Participants

This was an exploratory retrospective monocentric study. PwMS, previously enrolled in a cross-sectional observational study [8], attending the MS center at A.O.U. “Federico II”, were asked to give their consent to the use of observational clinical data collected during their follow-up visits for research purposes. Of the 59 PwMS originally enrolled, 17 subjects gave their consent and were enrolled in the present follow-up study. The study was performed in accordance with good clinical practices and the Declaration of Helsinki. Inclusion criteria were: age > 12 years, onset < 25 years and diagnosis of relapsing remitting MS according to the 2010 McDonald′s criteria [11].

Exclusion criteria were ongoing clinical relapse and steroids in the last 30 days before questionnaire filling and psychiatric illnesses.

During follow-up visits, scheduled as for clinical practice (mostly every three months), the following information was collected: clinical disability assessed via Expanded Disability Status Scale (EDSS) and current disease-modifying treatment (DMT). Since clinical and psychological assessments were part of the clinical practice and all patients signed an informed consent form for the use of observational clinical data for research purposes, specific ethics approval was not required. At baseline and at year 4, patients were asked to fill the Pediatric Quality of life inventory (PedsQL) version 4.0 [12], an instrument characterized by high internal reliability (alpha = 90). Two psychologists administered the PedsQL. This questionnaire consists of 23 items, divided in four subscales: physical functioning (8 items, e.g., “I have difficulty walking more than 100 m”); emotional functioning (5 items, e.g., “I am frightened or afraid”); social functioning (4 items; e.g., “It is difficult for me to get along with other boys and girls”); and school/work functioning (5 items, e.g., “It is hard for me to pay attention to the lesson”). Every item is linked to a 5-point Likert scale, with the subject indicating how often the specific situation happens from 0 (never) to 4 (almost always). For scoring purposes, each value was converted into a linear scale with values from 0 to 100 (0 = 100, 1 = 75, 2 = 50, 3 = 25, and 4 = 0), so that higher scores would correspond to better quality of life. An overall score was computed as the sum of all the items, while sub-scores for the physical, emotional, social and school functioning scales were computed as the sums of the relative items. A sub-score for the psychological functioning was obtained as the sum of the emotional, social and school functioning items. The questionnaire administration was followed by a non-structured interview aimed at discussing patients′ answers, to obtain a better insight of the patients′ emotional status and offer additional hints for result interpretation. The Italian version of the PedsQL has already been validated with an acceptable reliability rate [13]. Since then, it has also been used in both cross-sectional frameworks from our group [8] and longitudinal multicentric studies [14], confirming its applicability in the pediatric MS population.

### 2.2. Statistical Analyses

Longitudinal within-group comparison of HRQoL total score and sub-scores was performed via the Wilcoxon signed-rank test. As effect size for the Wilcoxon test, we reported the r value, calculated as Z statistic divided by the square root of the sample size. To investigate the effect of disease activity (relapses) on the HRQoL changes over time (only for the variables that showed a significant variation from baseline to follow-up), we employed a mixed-effects quantile model using the lqmm Rpackage [15], fixing the quantile level to the median. *P*-values and estimated 95% confidence intervals were obtained with bootstrapping. As random effect, the model had intercepts for subjects to consider the non-independence that stems from having double measures (baseline and follow-up) by the same subject. As fixed effects, the model contained the time variable (interval between baseline and follow-up) and the group variable to test the difference between patients experiencing relapses and clinically stable patients. The interaction term between time and group was also tested.

Statistical analysis was conducted in R (R Core Team, 2018). Statistical significance was set to α = 0.05 for all statistical tests

## 3. Results

### Clinico-Demographic Features

Demographic and clinical data are presented in Table 1. Over the follow-up, nine patients experienced clinical relapses and one patient showed disability progression (1 point increase in EDSS); twelve patients switched therapy (either for the presence of disease activity *n* = 6, or for reduced compliance to the DMT side effects *n* = 6). No longitudinal changes were observed in EDSS, in the overall PedsQL score, nor in the physical, school and psychological functioning (see Table 2). Conversely, an increase in the social functioning subscale, 45(30) vs. 95(25), r = 0.88 and *p* < 0.001, as well as a decrease in the emotional subscale, 80(25) vs. 50(30), r = 0.62 and *p* = 0.011, were observed (see Table 2).

For these two latter variables, we applied a mixed-effects quantile model and found that only the main effect of “time” was statistically significant for both social functioning (b_1_ = 37.9, 95% C.I. [26.6; 49.3] and *p* < 0.001) and emotional functioning (b_1_ = −25, 95% C.I. [−43.6; −6.4] and *p* = 0.008). There was a significant interaction between relapse occurrence and time in the model for social functioning (*p* = 0.032), while this interaction was not significant in the model for emotional functioning (*p* = 0.396). Figure 1 illustrates the medians and the first and third quartiles of each condition for social functioning (A) and emotional functioning (B).

Finally, we explored the impact of therapy switch on social and emotional functioning. Specifically, we applied a mixed-effects quantile model also considering the switch and found that the main effect of switch and the interaction between switch and time were not statistically significant in both the model for social functioning and the model for emotional functioning.

All values are expressed as median (IQR); the r value is the calculated effect size for the Wilcoxon signed-rank test.

## 4. Discussion

In this longitudinal evaluation, we examined HRQoL in early-onset PwMS over a 4-year period. While no variation in the global score was recorded, we identified a significant increase over time in social functioning and, conversely, a significant decrease in emotional functioning. Social functioning improvement over time was significantly affected by the occurrence of relapses.

The stability of the QoL global score over time is consistent with reports in adult PwMS, who also showed stable or improved social and emotional sub-scores over time [16]. Conversely, in our population of early-onset MS patients, this holds true only for social functioning [4,16,17]. Indeed, social support and a functional family environment contribute to maintaining a favorable psychological status, notwithstanding the presence of disease [18,19].

Social relationships seem to be a resource in daily adolescent life [20]. Friendly relationships give adolescents practical and emotional support. Peer confrontation, as a matter of fact, makes them feel supported in daily activities and lets patients regain a sense of normalcy [21]. In our population, friends were reported to help with schoolwork, carrying bags, raising spirits, listening, advocating and making allowances for physical limitations. Young people may reframe their sense of normalcy also by using downward social comparisons to define themselves as fortunate as a means of adjusting to a sense of difference [8,22]. Better social functioning would increase the patients’ chances to face disease limitations in daily life through the interaction with peers and others in general. Indeed, with increasing time intervals from disease diagnosis, an assimilation and adaptation to the condition of disease occurs, which facilitates openness to social interactions and external support [23]. In our population, social functioning improvement over time was slightly more evident in patients experiencing relapses, a finding that might point to a tendency to turn towards others for support in young patients with higher disease activity.

From an emotional standpoint, a diagnosis of a chronic and degenerative illness such as MS during early life represents an individual and family psychosocial experience of suffering, amplified by the feelings of fear and depression linked to the illness, ambivalence towards one’s family (“I am angry with my family/I need my family”) and the presence of an identity conflict. In fact, particularly during adolescence, identity and the integration of the experiences of a changing body into a coherent self-awareness are still developing. The complexity of these emotional experiences can hinder the developmental process, hampering resilience and resulting in a lack of acceptance of the disease and worsening of quality of life [24].

Patients with POMS are also at a significantly increased risk for depression, fatigue and loss of HRQoL. Furthermore, fatigue and depression were found to significantly predict reduced HRQoL in POMS [25].

In our population, while no significant changes in total HRQoL were noted, the patients’ emotional functioning worsened over time. Emotional experiences such as fear, sadness, anger and difficulty sleeping and, above all, worries about future and the disease progression, were frequently reported by the PwMS of our sample, mostly related to the perspective of living with a chronic, life-long neurodegenerative condition. This seems to be due to a greater actual awareness of their own health condition compared with the earlier disease course, which is also related to the adult age. Disease and therapy experience led the subjects to face the limitations caused not only by their condition but also by fantasies and ghosts linked to it, which might explain the worsening of the emotional domain observed over time.

Variations in emotional and social functioning over time were not affected by DMT changes, even though the therapeutic switch, in all cases, determined the substitution of an injectable with an oral/infusible drug. Such switches result in an improvement in therapy adherence and persistence related to a greater acceptability of the route of administration [26] but likely do not have an impact on HRQoL changes that are an expression of the patients’ mechanisms of coping with the disease and its evolution.

Recently, a very interesting paper by O′Mahony J and colleagues on HRQoL trajectories over time in MS, and the related risk factors, was published. The very large sample of adult PwMS, with a mean age of 41.7 (9.5) years at diagnosis, showed that older age at diagnosis, worse physical impairments and worse fatigue were associated with increased odds of being in the group with the worst physical HRQoL [27]. According to their results, patients with early MS onset should be “protected” by HRQoL worsening, as in our small sample. However, EDSS stability and the absence of fatigue scales in our study do not allow comparing their results with our results from a smaller sample.

We must declare that the small number of pediatric PwMS and the exploratory retrospective nature of the study is a limitation of our study, but this is in line with the rarity of pediatric onset of MS in a monocentric study context. Moreover, the wide age range, from childhood to early adulthood, might mine the analysis of influencing variables; however, we used the appropriate tool versions according to age. In spite of these limits, our results are relevant to inform proper psychological interventions in adolescents and young adults with MS. Particularly, it appears fundamental to boost patients’ social resources while simultaneously supporting them to keep in touch with health-related emotional aspects.

Indeed, stimulating the patients to accept their emotional responses to health-related limitations, while preserving their social and relational resources, seems key to the preservation of an adequate QoL over time. In particular, a recent review highlighted that an aspect that may reduce emotional distress in children and adolescents with MS is encouraging their own involvement during the diagnosis and decision-making process, as well as the relevance of an increased educational support by means of both increased awareness and disease specific accommodations [28].

## 5. Conclusions

In conclusion, as a priority, we suggest that psychological interventions in adolescents and young adults with MS should be structured with the aim of promoting the integration between the emotional aspects and the social resources, since emotional functioning seems to be at risk in young PwMS; on the other hand, social functioning in young patients is more effective, in particular when clinical activity is higher and there can be more “caring” figures around them.

## Figures and Tables

**Figure 1 jcm-11-05226-f001:**
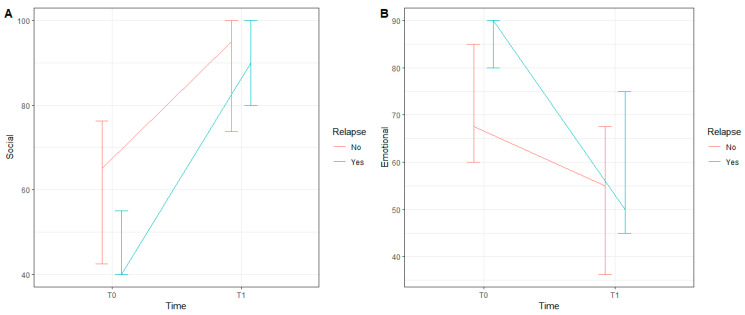
Median, first and third quartiles of each condition for Social Functioning (**A**) and Emotional Functioning (**B**).

**Table 1 jcm-11-05226-t001:** Demographic and clinical characteristics of the study population. EDSS: Expanded Disability Status Scale.

Subjects	17
Female sex, *n* (%)	11 (64.7%)
Age, mean ± SD (years)	24 ± 3.2
Age at onset, mean ± SD (years)	17 ± 2.9
Disease duration at baseline, mean ± SD (years)	1.70 ± 1.16
EDSS median (range)	2.5 (1.5–4)

**Table 2 jcm-11-05226-t002:** Longitudinal changes in clinical scales. PedsQL: Pediatric Quality of Life Inventory; EDSS: Expanded Disability Status Scale.

	T0	T1	r	*p*-Value
Pediatric Quality of Life Inventory (PedsQL) Total score	66.3 (27.2)	78.3 (31.5)	0.34	0.170
School Functioning	65.6 (28.7)	70 (30)	0.02	0.943
Social Functioning	45 (30)	95 (25)	0.88	<0.001
Emotional Functioning	80 (25)	50 (30)	0.62	0.011
Physical Functioning	70 (40)	71.9 (40.6)	0.36	0.142
Psychological Functioning	62.1 (28.3)	75 (29.7)	0.19	0.459
EDSS	2.5 (1.5–4)	2.5 (1.5–4)		1

## Data Availability

Data will be made available upon reasonable request to the corresponding author.

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
