# Peer review of "Quality of Life Changes in Early-Onset Multiple Sclerosis: A 4-Year Follow-Up Study"

_jcm, 2022, doi:10.3390/jcm11175226_

Round 1

Reviewer 1 Report

In their article the authors describe the effects of pediatric MS on health quality issues overtime using the PedQL at baseline and 4 years later. The overall scores remained stable overtime but social functioning which was influenced by the number of relapses and emotional functioning did change overtime.

The article is in most parts well written and addresses important points in the field of pediatric MS.

The references could be substituted by two publications by Storm Van's Gravesande and colleagues: Fatigue and depression predict health-related quality of life in patients with pediatric-onset multiple sclerosis. Mult Scler Relat Disord. 2019 Nov;36:101368 and Storm Van's Gravesande: The Multiple Sclerosis Inventory of Cognition for Adolescents (MUSICADO): A brief screening instrument to assess cognitive dysfunction, fatigue and loss of health-related quality of life in pediatric-onset multiple sclerosis. Eur J Paediatr Neurol. 2019 Nov;23(6):792-800. 

My main concern is the small number of children included in this study and the wide age range of 12 to 25 years certainly influencing variables such as social and emotional functioning. From a pediatric point of view the focus should have been on the group >12-18. 

Author Response

In their article the authors describe the effects of pediatric MS on health quality issues overtime using the PedQL at baseline and 4 years later. The overall scores remained stable overtime but social functioning which was influenced by the number of relapses and emotional functioning did change overtime.

The article is in most parts well written and addresses important points in the field of pediatric MS.

We thank the reviewer for having thoroughly revised our work providing valuable suggestions.

The references could be substituted by two publications by Storm Van's Gravesande and colleagues: Fatigue and depression predict health-related quality of life in patients with pediatric-onset multiple sclerosis. Mult Scler Relat Disord. 2019 Nov;36:101368 and Storm Van's Gravesande: The Multiple Sclerosis Inventory of Cognition for Adolescents (MUSICADO): A brief screening instrument to assess cognitive dysfunction, fatigue and loss of health-related quality of life in pediatric-onset multiple sclerosis. Eur J Paediatr Neurol. 2019 Nov;23(6):792-800. 

We added the proposed references accordingly

My main concern is the small number of children included in this study and the wide age range of 12 to 25 years certainly influencing variables such as social and emotional functioning. From a pediatric point of view the focus should have been on the group >12-18. 

We agree that the number of pediatric patients is limited, and we are discussing this point more extensively, however the focus is on early onset MS (i.e onset before 26 years of age), and we used the appropriate versions of the questionnaires accordingly.

We thank the reviewer for letting us further elaborating on the limitations of the present paper. We improved the limitation section in discussion accordingly.

Reviewer 2 Report

1. The introduction should be extended - it's too cursory.

A number of existing articles have not been taken into account by the authors. Examples:

https://pubmed.ncbi.nlm.nih.gov/26236506/

https://www.ncbi.nlm.nih.gov/pmc/articles/PMC7980344/

https://www.ncbi.nlm.nih.gov/pmc/articles/PMC4831037/

https://www.ncbi.nlm.nih.gov/pmc/articles/PMC3883029/

These are just examples, there are quite a few of them. The article is not innovative.

2. Reliability has not been calculated for all scales / questionnaires used.

3. The exclusion criterion has not been described.

4. For such a small sample size, the applied statistical tests are incorrect.

5. Statistical test results are not recorded according to scientific standards.

6. The effect size for the tests used has not been calculated and interpreted. The p-value alone is not enough.

7. The power of the used test has not been tested.

8. The number of analyzes carried out is negligible.

9. The type of descriptive statistics used is not appropriate for this sample size.

10. The discussion is based on an incorrectly conducted analysis.

11. The limitations have been described very briefly.

Author Response

We thank Reviewer 2 for the meticulous revision , that has pushed us to extensively improve our paper, especially with reference to statistical methods and results review. We hope to meet requested amendments for each raised concern appropriately.

  1. The introduction should be extended - it's too cursory.

We improved the introduction trying not to be redundant on the well known QoL topic.

A number of existing articles have not been taken into account by the authors. Examples:

https://pubmed.ncbi.nlm.nih.gov/26236506/

https://www.ncbi.nlm.nih.gov/pmc/articles/PMC7980344/

https://www.ncbi.nlm.nih.gov/pmc/articles/PMC4831037/

https://www.ncbi.nlm.nih.gov/pmc/articles/PMC3883029/

These are just examples, there are quite a few of them. The article is not innovative.

We agree that literature on HRQoL and MS is extensive, there are hundreds of related papers, but our aim was not to perform a systematical review of literature. Our aim was to study the evolution of HRQoL in a sample of young patients with MS over time, analyzing more in depth emotional variables, that is still lacking in literature. Following your kind suggestion, we are including in the manuscript a reference to the most recent and highly representative paper by the group of Ruth Ann Marrie (O'Mahony J, Salter A, Ciftci B, Fox RJ, Cutter GR, Marrie RA. Physical and Mental Health-Related Quality of Life Trajectories Among People With Multiple Sclerosis. Neurology. 2022 Aug 10:10.1212/WNL.0000000000200931. doi: 10.1212/WNL.0000000000200931. Epub ahead of print. PMID: 35948450.) on QoL trajectories over time. In this paper, authors highlighted that there are very few longitudinal papers on this topic and there is an urgent need for studies aiming at evaluating QoL longitudinally, especially from the very early disease stages. Our article aims at providing information on this topic and, hence, we believe it would be useful and additive to the available literature.

  1. Reliability has not been calculated for all scales / questionnaires used.

The Italian version of the PedsQLhas been already validated with a acceptable reliability rate (Trapanotto et al. 2009). Since then it has also been used in both cros-sectional framework from our group (Lanzillo et al., 2016) and longitudinal multicentric studies (Ghezzi et al., 2017) confirming its applicability in pediatric MS population. To confirm PedsQoL validity and reliability was not the scope of the present paper and does not add novelty to available evidence. We included in the methods section a sentence outlining PedsQoL reliability and validity.

  1. The exclusion criterion has not been described.

Thank you for pointing to this missing information. We have now included exclusion criterion according to reviewer’s valuable suggestion.

  1. For such a small sample size, the applied statistical tests are incorrect.

Because variable distributions were not skewed and not characterized by outliers, we used parametric tests for our statistical analysis. However, given the legitimate reviewer's concern on this point, we have now used non-parametric statistical tests for all statistical analysis. The Wilcoxon signed-rank test was used for the longitudinal within group comparison and then we ran quantile mixed effects models (fixing the quantile level to the median) with a random intercept for subjects, and time and groups as fixed effects. It was tested also the interaction term between time and group. Please find amended statistical methods and reults according to our new analyses.

  1. Statistical test results are not recorded according to scientific standards.

According to the statistical tests and methods that we used in the revised version of the paper, we provided median and interquartile range (IQR) as descriptive statistics, p-values for the Wilcoxon signed-rank test, and beta coefficients, p-values and corresponding estimated 95% confidence intervals for quantile mixed effects model results. 

  1. The effect size for the tests used has not been calculated and interpreted. The p-value alone is not enough.

We have reported now the r value, calculated as Z statistic divided by the square root of the sample size, as effect size for the Wilcoxon test. As regards to the quantile mixed effects models, we reported the beta coefficients with the corresponding estimated 95% confidence intervals, which are not standardized effect sizes but give an insight of the magnitude of the effect, as they represent the difference between the estimated conditional median.

  1. The power of the used test has not been tested.

We acknowledge that the power of the used test is surely low, but this work was positioned as an exploratory study, because it is the first study investigating the evolution of HRQoL in a sample of early onset MS (i.e onset before 26 years of age) . Therefore, an a priori power analysis was not performed. We outpointed this issue in the methods and the limitation section. Results from this study will be helpful for the design of new studies, where power analysis will be performed a priori.

  1. The number of analyzes carried out is negligible.

We thank the reviewer for raising this point. We have now modified statistical analysis methods as to be more accurate and we have also modified the results section accordingly.

  1. The type of descriptive statistics used is not appropriate for this sample size.

As abovementioned, according to the statistical tests and methods we used in the revised version of the paper, we have used now median and interquartile range (IQR) as descriptive statistics.

  1. The discussion is based on an incorrectly conducted analysis.

According to reviewer’s suggestion we have now revised the whole statistical methods and we have reported results accordingly. We did not find substantial changes in our results. However we slightly modified discussion section as to tone down the whole findings and to be more consistent to our results.

  1. The limitations have been described very briefly.

We thank the reviewer for letting us further elaborating on the limitations of the present paper. We improved the limitation section in discussion accordingly.

Round 2

Reviewer 1 Report

thanks for the much improved manuscript

Reviewer 2 Report

The recommended changes have been incorporated.